# Wealth Distribution Involving Psychological Traits and Non-Maxwellian Collision Kernel

**DOI:** 10.3390/e27010064

**Published:** 2025-01-12

**Authors:** Daixin Wang, Shaoyong Lai

**Affiliations:** School of Mathematics, Southwestern University of Finance and Economics, Chengdu 611130, China; laishaoy@swufe.edu.cn

**Keywords:** wealth distribution, value function, non-Maxwellian collision kernel, Boltzmann equation

## Abstract

A kinetic exchange model is developed to investigate wealth distribution in a market. The model incorporates a value function that captures the agents’ psychological traits, governing their wealth allocation based on behavioral responses to perceived potential losses and returns. To account for the impact of transaction frequency on wealth dynamics, a non-Maxwellian collision kernel is introduced. Applying quasi-invariant limits and Boltzmann-type equations, a Fokker–Planck equation is derived. We obtain an entropy explicit stationary solution that exhibits exponential convergence to a lognormal wealth distribution. Numerical experiments support the theoretical insights and highlight the model’s significance in understanding wealth distribution.

## 1. Introduction

The investigation of wealth distribution has been carried out through the application of kinetic and statistical models, which provide a robust theoretical framework for analyzing complex socioeconomic phenomena. These models, fundamentally grounded in probability theory and statistical mechanics, have proven to be useful in elucidating the underlying mechanisms of wealth distribution [1], opinion formation [2], and knowledge accumulation [3]. In particular, by simulating interactions between individual agents, researchers have been able to replicate and analyze the emergence of power-law tails and other complex features in wealth distribution [4].

Bouchaud and Mezard [5] introduced the collision operator (Fokker–Planck operator) to describe the process of wealth exchange. Chakraborti and Chatterjee [1] developed wealth exchange models by using the theory of rarefied gas dynamics to study wealth distribution problems. Cordier et al. [4] established a rule for wealth exchange based on the binary interaction followed by gas molecule collisions, integrating economic market behavior into the model. They applied a scale transformation to derive the Fokker–Planck operator. The results in [4] indicate that the density function of the wealth distribution satisfies a Boltzmann equation, and the steady-state solution of its Fokker–Planck equation is an inverse Gamma function, implying the presence of a Pareto tail in the steady-state wealth distribution. Bertotti [6] considered a model of the taxation and redistribution process in a closed society. Toscani [2] presented a kinetic model of continuous opinion formation, accounting for both opinion exchange between agents and information diffusion. Düring et al. [7] used a Boltzmann-type control model to analyze how leaders adjust their strategies based on an objective functional to achieve opinion consensus, leveraging an instantaneous binary control formulation that integrates cost minimization into their interactions. From the Boltzmann-type equation, the Fokker–Planck equation is derived, which gives explicit stationary solutions. Pareschi and Toscani [3] proposed a nonlinear Boltzmann-type kinetic equation, incorporating trade parameters such as risk awareness and saving propensity decided by the individual knowledge of the agent. This model aims to explore how variations in knowledge levels influence the evolution of wealth distribution among agents. Numerical simulations illustrate that high levels of knowledge can give rise to a wealthy class and exacerbate wealth inequality.

Recent developments have emphasized the incorporation of behavioral economic principles, particularly the integration of individual characteristic variables (such as beliefs and reliabilities), which are embedded in the agents’ strategies in microdynamic wealth exchange models, thereby making the model more realistic. Specifically, Brugna and Toscani [8] explored a nonlinear Boltzmann-type kinetic equation that incorporates the impact of individual conviction on opinion formation among agents, modifying binary exchanges by allowing the compromise and diffusion terms to depend on conviction levels. Numerical simulations reveal that conviction can break symmetry and generate opinion clusters. Toscani et al. [9] investigated wealth distribution by employing a value function to model the psychological behavior of agents engaged in strategic interactions, indicating that the wealth distribution of the agents adheres to a lognormal distribution. In the investigation of wealth distribution within financial markets, Hu and Lai [10] analyzed how variations in knowledge levels affect wealth allocation in the stock market. It was found in [10] that the psychological traits of agents could be effectively described using a value function that incorporates both the present price of stock and its essential value. Zhong et al. [11] considered a value function relating to agents’ risk perception behavior, which characterizes the proportion of wealth allocation between risky and riskless assets. By integrating this value function into the interaction rules between stocks and bonds, the authors in [11] examined the micro-level variations in investment strategies influenced by price and risk factors, concluding that the wealth invested in stocks also converges to a lognormal distribution.

In addition, in the context of wealth distribution models based on Boltzmann dynamics, the traditional approach utilizes a Maxwellian collision kernel, which presumes a constant interaction frequency among agents engaged in market activities. This reflects the assumption that agents interact uniformly, regardless of their wealth, implying a sort of ‘egalitarian’ trading behavior in the system. In contrast, non-Maxwellian collision kernels (such as power and linear collision kernels) allow for more realistic interaction patterns that take the agents’ wealth into account.

Furioli et al. [12] proposed a non-Maxwellian collision kernel incorporated into the Boltzmann collision integral operator, implying that the interaction frequency between agents varies, rather than remaining constant. Additionally, an update rule for the evolution of agent wealth is proposed in [12], where a Fokker–Planck equation is derived from the Boltzmann equation and the corresponding wealth distribution is analyzed.

Recently, scholars have highlighted the effectiveness of kinetic models in capturing the nuances of asset trading and wealth evolution. Building upon these foundations, particularly the work in [11], in this paper, we focus on a financial market characterized by the presence of both a risky asset (such as a stock) and a riskless asset (like a bond). We develop a kinetic exchange model to explore how wealth is distributed among agents engaged in speculative trading and wealth exchanges. Our contributions are central to three points, as follows.

(i) The value function used to describe the asset allocation of agent in Zhong et al. [11] is$$\Psi \left(\frac{X(t)}{\tilde{X}\left(t\right)}\right)=\frac{{\left(\frac{X(t)}{\tilde{X}\left(t\right)}\right)}^{\delta }-1}{{\left(\frac{X(t)}{\tilde{X}\left(t\right)}\right)}^{\delta }+1},$$
where δ∈(0,1) is a constant, and X(t) and $\tilde{X}\left(t\right)$ are the current price and intrinsic price of the stock, respectively. We choose a different value function to illustrate the feature of binary interaction dynamics based on the evolution that agents shift wealth between two assets, which reads as$$\Psi \left(\frac{X(t)}{\tilde{X}\left(t\right)}\right)=\frac{{\left(\frac{X(t)}{\tilde{X}\left(t\right)}\right)}^{\delta }-1}{{\left(\frac{X(t)}{\tilde{X}\left(t\right)}\right)}^{\delta }+1-\beta },$$
where β∈(0,1) is a key parameter in modeling the behavior of agents, capturing the influence of psychological characteristics during trading. This value function reflects agents’ behavioral responses to potential returns and losses, depending not only on the intrinsic price and current price of the risky asset but also on the parameter β. In the value function $\Psi (\frac{X(t)}{\tilde{X}\left(t\right)})$, parameter β∈(0,1) is explained as a parameter of psychological traits that depict the agents’ strategies, which are affected by psychological tendencies during trading. Specifically, the introduction of β causes the value function $\Psi (\frac{X(t)}{\tilde{X}\left(t\right)})$ to approach the reference point $\Psi (\frac{X(t)}{\tilde{X}\left(t\right)})=1$ more gradually when the current stock price exceeds its intrinsic price, as seen in Figure 1. This setting suggests that agents slowly adjust their portfolios, showing reluctance to reduce stock holdings in response to elevated prices of risky assets, which reflects their psychological traits in profit maximization under such market conditions. This behavior aligns with the trading patterns commonly observed in financial markets.

(ii) Inspired by the work in [12], we apply a non-Maxwellian collision kernel relating with the frequency of wealth exchange to consider the effect of the transaction frequency among agents. By contrast with the work in Zhong et al. [11] where the constant collision kernel was used, we analyze the impact of agents’ transaction frequency on the evolution of wealth distribution.

(iii) The stationary solution we obtained is different from those in Zhong et al. [11]. Our stationary solution is related with parameter of transaction frequency and parameter of psychological traits, which was not considered in [11].

To validate our theoretical results, we conduct numerical experiments that illustrate the dynamics of wealth distribution under various scenarios. These experiments not only confirm the robustness of our model but also provide deep insights into the factors that affect the long-term behavior of wealth distribution in financial markets.

This paper is structured as follows: Section 2 describes the construction of the kinetic exchange model and the incorporation of the value function. Section 3 discusses the main properties of the kinetic equation. Section 4 presents the derivation of the Fokker–Planck equation and the existence of a stationary solution. Section 5 offers a detailed analysis of the numerical experiments, while Section 6 concludes with a summary of our results.

## 2. A Kinetic Description of Wealth Distribution

In essence, the kinetic model of wealth exchange in our paper is inspired by the model in Zhong et al. [11]. We consider that agents in the market equipped with a risky asset (stock) and a risk-free asset (bond) as their wealth, aiming to describe the evolution of wealth distribution. We assume that the agents are homogeneous, with non-negative wealth, and that the allocation of such assets can be adjusted at any point in time. Thus, we depict the dynamics of wealth distribution among agents via the interplay of the two kinds of assets.

Considering the microscopic interplay of the risky and risk-free assets (stocks and bonds), we apply v≥0 to denote each agent’s wealth invested in stocks, and use w≥0 to denote the rest part invested in bonds. Furthermore, companying agents trading with others in the market, an agent’s wealth before interaction (v,w) is adjusted to the post-interaction $\left(v^{*},w^{*}\right)$. Then, we depict the interaction rule as follows(1)$$\begin{array}{c} \left\{\begin{array}{c} v^{*}=p_{1}v+q_{1}w,\\ w^{*}=q_{2}w+p_{2}v, \end{array}\right. \end{array}$$
where the non-negative coefficients $p_{i}$ and $q_{i}$(i=1,2) are variables related to the stock’s price.

Generally, portfolio management involves making choices. As agents continuously adjust their investment strategies based on predictions of future market conditions, agents determine the proportion of their investments in both risky and risk-free assets at each stage of the process. Consequently, the price dynamics of the risky asset are affected by the portfolio decisions, which are made by agents. We assume that the profit rate of the risk-free asset is a fixed interest rate r≥0, and the profit rate of the risky asset is described by$$\frac{\dot{X}\left(t\right)+D}{X(t)},$$
where X(t) is the stock price at time *t*, D(t)≥0 is the dividend, and $\dot{X}\left(t\right)$ denotes the time derivative of X(t). We define the current price of the stock by using the formulation in [11], which reads as(2)$$X\left(t\right)=X\left(0\right)e^{\nu P(t)t},$$
where P(t) represents the average investment propensity of agents, which satisfies −1<P(t)<1, and parameter ν>0 indicates the rate of price evolution. Equation (Equation 2) shows that the current stock price is related to the average investment propensity of agents, which is based on the idea in [13].

In fact, comparing the return from the investment of bond, agents strategically adjust their stock holdings in the next step to maximize their profits. We define the stock’s excess profit rate which is denoted by χ as follows(3)$$\chi =\frac{X_{p}\left(t\right)-X\left(t\right)+D}{X(t)}-r,$$
where $X_{p}\left(t\right)$ represents the predictable stock price. Equation (Equation 3) implies the potential gain or loss comparing with investing in bonds.

Note that, when the profit rate of the investment in risky assets is equal to the profit rate of risk-free assets, the stock’s intrinsic price, which is denoted by $\tilde{X}\left(t\right)$, satisfies(4)$$\tilde{X}\left(t\right)=\frac{X_{p}\left(t\right)+D}{1+r}.$$

We infer from Equation (Equation 4) that the intrinsic price of the stock relating to the stock’s predictable price influences the excess profits of the agent, and consequently affects the agent’s investment decision.

According to the prospect theory in [14,15], agents formulate and carry out their investment strategy to get excess profit by identifying the gap between the stock’s current price and intrinsic price. To be specific, when$$\frac{X_{p}\left(t\right)-X\left(t\right)+D}{X(t)}-r>0,$$
we have$$\tilde{X}\left(t\right)=\frac{X_{p}\left(t\right)+D}{1+r}>X\left(t\right).$$ The fixed interest rate of the bond is below the profit rate of the stock, leading to a positive excess profit rate that compels agents to shift their wealth from the bond to the other asset. Conversely, when$$\frac{X_{p}\left(t\right)-X\left(t\right)+D}{X(t)}-r<0,$$
we have$$\tilde{X}\left(t\right)=\frac{X_{p}\left(t\right)+D}{1+r}<X\left(t\right).$$ Owing to χ<0, the investment in the bond has relatively higher yields, and so agents would like to make a reduction gradually on their investment in stocks.

In order to investigate the exchanges of wealth, we introduce a value function Ψ which satisfies(5)$$p_{i}=1-\gamma \Psi \left(z\right)+\eta_{i}\;\left(i=1,2\right)$$
and$$q_{i}=\gamma \Psi \left(z\right)\;\left(i=1,2\right).$$ In Equation (Equation 5), *z* represents a positive variable, and $\eta_{i}\left(i=1,2\right)$ denotes random variables that capture market fluctuations, with zero mean and variance $\sigma^{2}$, while 0<γ<1 is a constant that quantifies the proportion of wealth transferred between assets. According to [4], it can be explained as a transaction coefficient.

By substituting $z=\frac{X(t)}{\tilde{X}\left(t\right)}$ into Equation (Equation 1), we derive the following binary interaction model, which characterizes the wealth distribution’s microscopic dynamics between the investments of risky and riskless assets, which reads$$\begin{array}{c} \left\{\begin{array}{c} v^{*}=v-\gamma \Psi \left(\frac{X(t)}{\tilde{X}\left(t\right)}\right)v+\gamma \Psi \left(\frac{X(t)}{\tilde{X}\left(t\right)}\right)w+\eta_{1}v,\\ \;w^{*}=w-\gamma \Psi \left(\frac{X(t)}{\tilde{X}\left(t\right)}\right)w+\gamma \Psi \left(\frac{X(t)}{\tilde{X}\left(t\right)}\right)v+\eta_{2}w. \end{array}\right. \end{array}$$
We choose the value function Ψ as$$\Psi \left(\frac{X(t)}{\tilde{X}\left(t\right)}\right)=\frac{{\left(\frac{X(t)}{\tilde{X}\left(t\right)}\right)}^{\delta }-1}{{\left(\frac{X(t)}{\tilde{X}\left(t\right)}\right)}^{\delta }+1-\beta },$$
where 0<β<1 represents an appropriate parameter to describe the behavior of the agent affected by psychological traits. The exponent δ∈(0,1) is a constant that controls how strongly the value function responds to the ratio between the current price and the intrinsic price of risky assets. Actually, we notice that $z=\frac{X(t)}{\tilde{X}\left(t\right)}>0$. Then, we find that the value function $\Psi (\frac{X(t)}{\tilde{X}\left(t\right)})$ is bounded, which satisfies$$-1<\Psi (\frac{X(t)}{\tilde{X}\left(t\right)})<1.$$
Figure 1 illustrates that the value function $\Psi (\frac{X(t)}{\tilde{X}\left(t\right)})$ represents the deviation from a reference point where $\frac{X(t)}{\tilde{X}\left(t\right)}=1$. Specifically, for $X\left(t\right)>\tilde{X}\left(t\right)$, the value function is positive, indicating risk-seeking behavior in response to losses. In contrast, for $X\left(t\right)<\tilde{X}\left(t\right)$, the function becomes negative and convex, reflecting risk aversion in the presence of gains.

Compared to the work in Zhong et al. [11], the primary distinction is observed on the right side of the reference point, where $\frac{X(t)}{\tilde{X}\left(t\right)}=1$. As shown in Figure 1, the curve of $\Psi (\frac{X(t)}{\tilde{X}\left(t\right)})$ is smoother than the left side. In this regard, the value function $\Psi (\frac{X(t)}{\tilde{X}\left(t\right)})$ approaches the reference point relatively slower when the current price of the stock exceeds the intrinsic price. This setup implies that agents gradually adjust their portfolios and sell stocks with hesitation in response to rising prices of risky assets, reflecting their psychological tendencies in profit maximization during such market conditions. This behavior is consistent with trading patterns frequently observed in financial markets.

## 3. The Kinetic Description of Wealth

In this part, we consider a Boltzmann dynamic according to the above binary interaction to derive a kinetic equation to study the evolution of wealth distribution. We introduce a density function h(v,t), where $v\subseteq R^{+}$ at time t>0. The specific meaning of the density *h* is as follows. Given the system of traders, and given an interval or a more complex sub-domain $D\subseteq R^{+}$, the integral$$\int_{D}h\left(v,t\right)dv.$$
represents the number of traders who are characterized by an amount of wealth v∈D at time t≥0. It is assumed that the density function is normalized to one; that is, for all t≥0$$\int_{R^{+}}h\left(v,t\right)dv=1.$$ Additionally, we suppose that the total amount of the stock owned by an agent, denoted by *n*, is fixed. Then, we use v=nX(t) to denote the current value of wealth aligned with the stock, while the expected value of wealth attributable to the stock under the predictability is represented by $v_{e}=n\tilde{X}\left(t\right)$. Thus, we obtain(6)$$\begin{array}{c} \left\{\begin{array}{c} v^{*}=v-\gamma \Psi \left(\frac{v}{v_{e}}\right)v+\gamma \Psi \left(\frac{v}{v_{e}}\right)w+\eta_{1}v,\\ \;w^{*}=w-\gamma \Psi \left(\frac{v}{v_{e}}\right)w+\gamma \Psi \left(\frac{v}{v_{e}}\right)v+\eta_{2}w. \end{array}\right. \end{array}$$ In rule (6), the value function $\Psi (\frac{v}{v_{e}})$ quantifies the variations relative to the reference point where $\left(\frac{v}{v_{e}}=1\right)$. The value function is positive above the reference point $\left(v>v_{e}\right)$ and negative below the reference point $\left(v<v_{e}\right)$.

Based on the interaction rule (6), the density function, which is denoted by h(v,t), satisfies the weak form of the Boltzmann-type equation. Specifically, for any smooth test function ρ(v) defined on $R^{+}$, the density function h(v,t) acts as the weak solution of the equation (7)$$\begin{array}{cc} \frac{d}{dt}\int_{R+}\rho \left(v\right)h\left(v,t\right)dw & =\frac{1}{2}\langle \underset{R+}{\;\int \int \;}\kappa v^{\alpha }\left(\rho \left(v^{*}\right)+\rho \left(w^{*}\right)-\rho \left(v\right)-\rho \left(w\right)\right)\\ \; & \;\;\;\;\;\;h(v,t)h(w,t)dvdw\rangle , \end{array}$$
where symbol 〈·〉 denotes the mathematical expectation, and $\kappa v^{\alpha }$ is the collision kernel related to the transaction frequency of agents with constant κ>0 and the parameter of transaction frequency α, where 0<α≤1.

Note that Equation (Equation 7) is derived from the kinetic equation as originally formulated in [16], where the collision kernel in the originated equation is a constant term. However, in our work, we introduce a non-Maxwellian collision kernel related to the frequency of wealth exchange in risky assets to account for the effect of transaction frequency among agents. Specifically, the introduction of transaction frequency parameter α is intended to depict the interaction frequency of agents in the kinetic equation. In this context, the non-Maxwellian collision kernel we introduced is the power collision kernel (expressed as $\kappa v^{\alpha }$, where α is the parameter of transaction frequency, κ is a constant, and *v* represents the wealth of the agent invested in risky assets), which indicates that agents who do not invest in risky assets are not involved in trading. It characterizes the frequency of stock transactions executed by the agent at each trading point in kinetic Equation (Equation 7). By contrast with the constant collision kernel in Zhong et al. [11], this is our innovation.

**Remark** **1.**
*In particular, the power collision kernel introduces a wealth-dependent interaction frequency, implying that the likelihood of interaction is no longer constant but rather varies based on the wealth of the agents. This models real-world phenomena where wealthier individuals may engage more frequently in trade or speculative activities, or where the willingness to trade might be influenced by wealth disparities between agents. This feature often observed in real-world economies but not captured by Maxwellian kernels.*


**Remark** **2.**
*Compared to the power collision kernel, the linear collision kernel introduces a more direct relationship between agents’ wealth investment and their trading behavior. This type of kernel implies that the frequency of interaction between agents is a linear function of their wealth, which is different from the assumption of independent interactions in the general framework. In this case, wealthy agents might engage in trade at a rate proportional to their wealth, leading to differential impacts on wealth distribution. The interaction dynamics can generate varying levels of inequality based on how agents invest and interact, and the introduction of a bilinear kinetic model enhances the system’s ability to account for disparities in wealth investment across groups.*


**Remark** **3.**
*While earlier kinetic models often assume random or uniform interactions (such as in the mean-field models), the non-Maxwellian kernel allow the kinetic model to capture more heterogeneity by varying the interaction frequency based on wealth. This is closer to agent-based models in economics, where agents have different wealth, psychological traits, and risk profiles, leading to more complex and often heterogeneous market dynamics. The approach of introducing a non-Maxwellian kernel also aligns with behavioral economics, which considers psychological factors, risk aversion, and other human behavioral aspects that affect economic decision-making. This incorporation of agents’ psychological traits into the wealth evolution process creates a more realistic representation of market behavior, as agents’ interactions are not purely determined by objective wealth maximization but also by subjective preferences, social influences, and expectations.*


To be more precise, investigating the temporal behavior of the wealth distribution is inherently tricky due to the nonlinearity of the interaction described in the interaction rule (6). Therefrom, we analyze the fundamental characteristics of the kinetic equation.

Thanks to the entropy weak form of the Boltzmann-type equation, we can leave out the time dependence in this distribution function. Setting ρ(v)=1, we obtain the conservation of wealth in the system, meaning that Equation (Equation 7) is conserved. Namely,$$\langle v^{*}+w^{*}\rangle =\left(v+\langle \eta_{1}\rangle v^{*}+w+\langle \eta_{2}\rangle w^{*}\right)=v+w,$$
where $\langle \eta_{1}\rangle =\langle \eta_{2}\rangle =0$.

Now, taking ρ(v)=v in (7), we have that the evolution of average value of wealth satisfies(8)$$\frac{d}{dt}m_{1}\left(t\right)=0,$$
where$$m_{1}\left(t\right)=\int_{R^{+}}vh\left(v,t\right)dv.$$ It follows from (8) that $m_{1}\left(t\right)=m$ is a positive constant. Taking $\rho \left(v\right)=v^{2}$ yields(9)$$\frac{d}{dt}\int_{R^{+}}v^{2}h\left(v,t\right)dv=\frac{1}{2}\langle \underset{R^{+}}{\;\int \int \;}\kappa v^{\alpha }\left({\left(v^{*}\right)}^{2}+{\left(w^{*}\right)}^{2}-v^{2}-w^{2}\right)h\left(v,t\right)h\left(w,t\right)dvdw\rangle .$$ We assume that the wealth *v* and *w* have the same density function *h*, which is given by$$\int_{R^{+}}h\left(v,t\right)dv=\int_{R^{+}}h\left(w,t\right)dw=1.$$ Utilizing the interaction rule (6), we obtain
$$\begin{array}{cc} \langle {\left(v^{*}\right)}^{2}+{\left(w^{*}\right)}^{2}-v^{2}-w^{2}\rangle & =-2\gamma \Psi \left(\frac{v}{v_{e}}\right)\left[{\left(v-w\right)}^{2}-\gamma \Psi \left(\frac{v}{v_{e}}\right)\left(v^{2}-2vw+w^{2}\right)\right]\\ \; & \;+{\langle \eta_{1}\rangle }^{2}v^{2}+{\langle \eta_{2}\rangle }^{2}w^{2}\\ \; & =-2\gamma \Psi \left(\frac{v}{v_{e}}\right)\left[\left(1-\gamma \Psi \left(\frac{v}{v_{e}}\right)\right){\left(v-w\right)}^{2}\right]\\ \; & \;+{\langle \eta_{1}\rangle }^{2}v^{2}+{\langle \eta_{2}\rangle }^{2}w^{2}. \end{array}$$ Since ${\langle \eta_{1}\rangle }^{2}={\langle \eta_{2}\rangle }^{2}=\sigma^{2}$, we have(10)$$\begin{array}{cc} \langle {\left(v^{*}\right)}^{2}+{\left(w^{*}\right)}^{2}-v^{2}-w^{2}\rangle & =-2\gamma \Psi \left(\frac{v}{v_{e}}\right)\left[\left(1-\gamma \Psi \left(\frac{v}{v_{e}}\right)\right){\left(v-w\right)}^{2}\right]\\ \; & \;+\sigma^{2}v^{2}+\sigma^{2}w^{2}. \end{array}$$ Substituting (Equation 10) into (Equation 9) gives rise to(11)$$\begin{array}{cc} \frac{d}{dt}\int_{R^{+}}v^{2}h\left(v,t\right)dv & =\gamma \Psi \left(\frac{v}{v_{e}}\right)\left(\gamma \Psi (\frac{v}{v_{e}})-1\right)[\int_{R^{+}}\kappa v^{\alpha }v^{2}h\left(v,t\right)dv\\ \; & +\underset{R^{+}}{\;\int \int \;}\kappa v^{\alpha }w^{2}h\left(v,t\right)h\left(w,t\right)dvdw\\ \; & -\underset{R^{+}}{\;\int \int \;}\kappa v^{\alpha }2vwh\left(v,t\right)h\left(w,t\right)dvdw]\\ \; & +\frac{1}{2}\sigma^{2}\int_{R^{+}}\kappa v^{\alpha }v^{2}h\left(v,t\right)dv\\ \; & +\frac{1}{2}\sigma^{2}\underset{R^{+}}{\;\int \int \;}\kappa v^{\alpha }w^{2}h\left(v,t\right)h\left(w,t\right)dvdw. \end{array}$$ Although (11) has no explicit solution, we consider the value function $\Psi (\frac{v}{v_{e}})=s$ and let$$\begin{array}{c} m_{\alpha }\left(t\right)=\int_{R^{+}}v^{\alpha }h\left(v,t\right)dv=\int_{R^{+}}w^{\alpha }h\left(w,t\right)dw. \end{array}$$ We have $$\begin{array}{cc} \frac{dm_{2}\left(t\right)}{dt} & =\gamma s\left(\gamma s-1\right)[\kappa \cdot m_{\alpha +2}\left(t\right)+\underset{R^{+}}{\;\int \int \;}\kappa v^{\alpha }w^{2}h\left(v,t\right)h\left(w,t\right)dvdw\\ \; & \;\;\;-2\kappa m\int_{R^{+}}v^{\alpha +1}h\left(v,t\right)dv]+\frac{1}{2}\sigma^{2}\kappa \cdot m_{\alpha +2}\left(t\right)\\ \; & \;\;\;+\frac{1}{2}\sigma^{2}\underset{R^{+}}{\;\int \int \;}\kappa v^{\alpha }w^{2}h\left(v,t\right)h\left(w,t\right)dvdw. \end{array}$$ Since $m_{1}\left(t\right)=m$ is positive constant, 0<α≤1 and (12)$$\begin{array}{cc} m_{\alpha }\left(t\right)=\int_{R^{+}}v^{\alpha }h\left(v,t\right)dv & =\int_{R^{+}}{\left(vh(v,t)\right)}^{\alpha }{\left(h\left(v,t\right)\right)}^{1-\alpha }dv\\ \; & \leq {\left[\int_{R^{+}}vh\left(v,t\right)dv\right]}^{\alpha }{\left[\int_{R^{+}}h\left(v,t\right)dv\right]}^{1-\alpha }\\ \; & =m_{1}^{\alpha }\left(t\right), \end{array}$$
utilizing inequality (12) yields $$m_{\alpha +1}\left(t\right)\leq m^{\alpha +1},$$
which derives $$\begin{array}{cc} \frac{dm_{2}\left(t\right)}{dt} & =2\kappa \gamma s\left(\gamma s-1\right)\left[m_{\alpha +2}\left(t\right)-m_{1}\left(t\right)\cdot m_{\alpha +1}\left(t\right)\right]+\frac{1}{2}\sigma^{2}\kappa \cdot m_{\alpha +2}\left(t\right)\\ \; & +\frac{1}{2}\sigma^{2}\kappa \cdot m_{2}\left(t\right)\cdot m_{\alpha }\left(t\right)\\ \; & \leq \sigma^{2}\kappa \cdot m^{\alpha +2}. \end{array}$$ Then, we have$$\frac{dm_{2}\left(t\right)}{dt}\leq \kappa \sigma^{2}m^{\alpha +2},$$
implying that the second moment $m_{2}\left(t\right)$ remains bounded.

## 4. Fokker–Planck Modeling

In fact, solving Equation (Equation 11) poses certain difficulties. Under this circumstance, numerical simulation is a general way to discuss the problem. In order to analyze the solution, we use the asymptotic limits of the Boltzmann-type equation to derive the Fokker–Planck equation. The core idea is to introduce an appropriate scaling rule to derive the Fokker–Planck equation and consequently obtain the steady-state solution.

We introduce a very small variable ϵ, where 0<ϵ<<1. Let$$t\to \frac{\tau }{\epsilon },\eta_{i}\to \sqrt{\epsilon }\eta_{i},\Psi \left(\frac{v}{v_{e}}\right)\to \Psi_{\epsilon }\left(\frac{v}{v_{e}}\right)=\frac{{\left(\frac{v}{v_{e}}\right)}^{\epsilon }-1}{{\left(\frac{v}{v_{e}}\right)}^{\epsilon }+1-\beta }.$$ Following the idea in [11], we set $M_{\epsilon }\left(v\right)$ as (13)$$M_{\epsilon }\left(v\right)=\frac{1}{\epsilon }\Psi_{\epsilon }\left(\frac{v}{v_{e}}\right).$$ For smooth functions ρ(v) with supported set in $R^{+}$, the corresponding distribution of wealth is denoted by $h^{*}\left(v,\tau \right)$ at time t>0. When ϵ→0, we have $$\begin{array}{cc} \frac{d}{d\tau }\int_{R^{+}}\rho \left(v\right)h^{*}\left(v,\tau \right)dv & =\frac{1}{2\epsilon }\langle \underset{R^{+}}{\;\int \int \;}\kappa v^{\alpha }\left(\rho \left(v^{*}\right)+\rho \left(w^{*}\right)-\rho \left(v\right)-\rho \left(w\right)\right)\\ \; & \;\;\;\;\;\;h^{*}\left(v,\tau \right)h^{*}\left(w,\tau \right)dvdw\rangle . \end{array}$$ Back to the properties of kinetic equation we discussed, we will derive the Fokker–Planck equation. We have$$\begin{array}{cc} \; & \langle v^{*}-v\rangle =\epsilon \gamma M_{\epsilon }\left(v\right)\left(w-v\right),\\ \; & \langle {\left(v^{*}-v\right)}^{2}\rangle =\epsilon^{2}\gamma^{2}M_{\epsilon }^{2}\left(v\right){\left(w-v\right)}^{2}+\sigma^{2}v^{2}. \end{array}$$ Using the second-order Taylor expansion of smooth function ρ(v) around *v*, we have $$\begin{array}{cc} \langle \rho \left(v^{*}\right)-\rho \left(v\right)\rangle & =\rho^{\prime }\left(v\right)\langle v^{*}-v\rangle +\frac{\rho^{\prime \prime }\left(v\right)}{2}\langle {\left(v^{*}-v\right)}^{2}\rangle \\ \; & \;\;+\frac{1}{6}\langle \rho^{\prime \prime \prime }\left(v+\zeta \left(v^{*}-v\right)\right){\left(v^{*}-v\right)}^{3}\rangle , \end{array}$$
where ζ∈(0,1). Thus, we obtain $$\begin{array}{cc} \langle \rho \left(v^{*}\right)-\rho \left(v\right)\rangle & =\rho^{\prime }\left(v\right)\epsilon \gamma M_{\epsilon }\left(v\right)\left(w-v\right)+\frac{\rho^{\prime \prime }\left(v\right)}{2}\sigma^{2}v^{2}\\ \; & \;\;+R_{\epsilon }\left(v\right), \end{array}$$
where $R_{\epsilon }\left(v\right)$ is the residual term, given by $$\begin{array}{cc} R_{\epsilon }\left(v\right) & =\frac{1}{2}\rho^{\prime \prime }\left(v\right)\epsilon^{2}\gamma^{2}M_{\epsilon }^{2}\left(v\right){\left(w-v\right)}^{2}\\ \; & \;\;\;\;+\frac{1}{6}\langle \rho^{\prime \prime \prime }\left(v+\zeta \left(v^{*}-v\right){\left(v^{*}-v\right)}^{3}\right)\rangle . \end{array}$$ Thus, we obtain the rescale equation which reads $$\begin{array}{cc} \frac{d}{d\tau }\int_{R^{+}}\rho \left(v\right)h^{*}\left(v,\tau \right)dw & =\underset{R^{+}}{\;\int \int \;}\kappa v^{\alpha }\left[\gamma M_{\epsilon }\left(v\right)\left(w-v\right)\rho^{\prime }\left(v\right)+\frac{\sigma^{2}}{2\epsilon }v^{2}\rho^{\prime \prime }\left(v\right)\right]\\ \; & \;\;\;\;\;\;h^{*}\left(v,\tau \right)h^{*}\left(w,\tau \right)dvdw+\frac{1}{\epsilon }R_{\epsilon }\left(\tau \right), \end{array}$$
in which $R_{\epsilon }\left(\tau \right)$ satisfies $$\begin{array}{cc} R_{\epsilon }\left(\tau \right) & =\underset{R^{+}}{\;\int \int \;}\kappa v^{\alpha }R_{\epsilon }\left(v\right)h^{*}\left(v,\tau \right)h^{*}\left(w,\tau \right)dvdw. \end{array}$$

As ϵ→0, the reminder term $\frac{1}{\epsilon }R_{\epsilon }\left(\tau \right)$ satisfies$$\lim_{\epsilon \to 0}\frac{1}{\epsilon }R_{\epsilon }\left(\tau \right)=0.$$ Therefore, we have the following equation as the residual term approaches zero, which is given by$$\begin{array}{cc} \frac{d}{d\tau }\int_{R^{+}}\rho \left(v\right)h^{*}\left(v,\tau \right)dv & =\underset{R^{+}}{\;\int \int \;}\kappa v^{\alpha }\left[\gamma M_{\epsilon }\left(v\right)\left(w-v\right)\rho^{\prime }\left(v\right)+\frac{\sigma^{2}}{2\epsilon }v^{2}\rho^{\prime \prime }\left(v\right)\right]\\ \; & \;\;\;\;\;\;\;\;h^{*}\left(v,\tau \right)h^{*}\left(w,\tau \right)dvdw. \end{array}$$ Letting $\sigma^{2}=\lambda \epsilon$, and assuming that both ϵ and σ are sufficiently small, we derive the weak form $$\begin{array}{cc} \frac{d}{d\tau }\int_{R^{+}}\rho \left(v\right)h^{*}\left(v,\tau \right)dv & =\underset{R^{+}}{\;\int \int \;}\kappa v^{\alpha }\left[\gamma M_{\epsilon }\left(v\right)\left(w-v\right)\rho^{\prime }\left(v\right)+\frac{\lambda }{2}v^{2}\rho^{\prime \prime }\left(v\right)\right]\\ \; & \;\;\;\;\;\;\;\;h^{*}\left(v,\tau \right)h^{*}\left(w,\tau \right)dvdw. \end{array}$$ Using (13) yields $$\lim_{\epsilon \to 0}M_{\epsilon }\left(v\right)=\lim_{\epsilon \to 0}\frac{1}{\epsilon }\left[{\left(\frac{v}{v_{e}}\right)}^{\epsilon }-1\right]\frac{1}{{\left(\frac{v}{v_{e}}\right)}^{\epsilon }+1-\beta }=\frac{1}{2-\beta }log\left(\frac{v}{v_{e}}\right).$$ Then, we obtain the equation$$\begin{array}{cc} \frac{d}{d\tau }\int_{R^{+}}\rho \left(v\right)h^{*}\left(v,\tau \right)dv & =\int_{R^{+}}\kappa v^{\alpha }\left(\frac{\gamma }{2-\beta }mlog\frac{v}{v_{e}}-\frac{\gamma }{2-\beta }vlog\frac{v}{v_{e}}\right)\rho^{\prime }\left(v\right)h^{*}\left(v,\tau \right)dv\\ \; & +\int_{R^{+}}\frac{\lambda }{2}\kappa v^{\alpha +2}\rho^{\prime \prime }\left(v\right)h^{*}\left(v,\tau \right)dv. \end{array}$$

Assuming that the boundary term resulting from the integral tends to zero as it approaches infinity, we consider that the boundary value of the density function is zero and obtain(14)$$\begin{array}{cc} \; & \int_{R^{+}}\kappa v^{\alpha }\left(\frac{\gamma }{2-\beta }mlog\frac{v}{v_{e}}-\frac{\gamma }{2-\beta }vlog\frac{v}{v_{e}}\right)\rho^{\prime }\left(v\right)h^{*}\left(v,\tau \right)dv\\ \; & =\int_{R^{+}}\kappa v^{\alpha }\left(\frac{\gamma }{2-\beta }mlog\frac{v}{v_{e}}-\frac{\gamma }{2-\beta }vlog\frac{v}{v_{e}}\right)\frac{d\rho (v)}{dv}h^{*}\left(v,\tau \right)dv\\ \; & =\left(\frac{\gamma \kappa }{2-\beta }mv^{\alpha }log\frac{v}{v_{e}}-\frac{\gamma \kappa }{2-\beta }v^{\alpha +1}log\frac{v}{v_{e}}\right)\rho \left(v\right)h^{*}\left(v,\tau \right){|}_{0}^{\infty }\\ \; & \;-\int_{R^{+}}\rho \left(v\right)\frac{d(\left(\frac{\gamma \kappa }{2-\beta }mv^{\alpha }log\frac{v}{v_{e}}-\frac{\gamma \kappa }{2-\beta }v^{\alpha +1}log\frac{v}{v_{e}}\right)h^{*}\left(v,\tau \right))}{dv}dv\\ \; & =-\int_{R^{+}}\rho \left(v\right)\frac{d(\left(\frac{\gamma \kappa }{2-\beta }mv^{\alpha }log\frac{v}{v_{e}}-\frac{\gamma \kappa }{2-\beta }v^{\alpha +1}log\frac{v}{v_{e}}\right)h^{*}\left(v,\tau \right))}{dv}dv \end{array}$$
and(15)$$\begin{array}{cc} \; & \int_{R^{+}}\frac{\lambda }{2}\kappa v^{\alpha +2}\rho^{\prime \prime }\left(v\right)h^{*}\left(v,\tau \right)dv\\ \; & =\int_{R^{+}}\frac{\kappa \lambda }{2}v^{\alpha +2}\frac{d\rho^{\prime }\left(v\right)}{dv}h^{*}\left(v,\tau \right)dv\\ \; & =\frac{\lambda \kappa }{2}v^{\alpha +2}\rho^{\prime }\left(v\right)h^{*}\left(v,\tau \right){|}_{0}^{\infty }-\frac{\lambda }{2}\int_{R^{+}}\rho^{\prime }\left(v\right)\frac{d(\kappa v^{\alpha +2}h^{*}\left(v,\tau \right))}{dv}dv\\ \; & =-\frac{\lambda }{2}\int_{R^{+}}\rho^{\prime }\left(v\right)\frac{d(\kappa v^{\alpha +2}h^{*}\left(v,\tau \right))}{dv}dv. \end{array}$$ From Equations (Equation 14) and (Equation 15), we derive the Fokker–Planck equation as follows (16)$$\frac{\partial h^{*}\left(v,\tau \right)}{\partial \tau }=-\frac{\partial [\left(\frac{\gamma \kappa }{2-\beta }mv^{\alpha }log\frac{v}{v_{e}}-\frac{\gamma \kappa }{2-\beta }v^{\alpha +1}log\frac{v}{v_{e}}\right)h^{*}\left(v,\tau \right)]}{\partial v}+\frac{\lambda }{2}\frac{\partial^{2}\left[\kappa v^{\alpha +2}h^{*}\left(v,\tau \right)\right]}{\partial v^{2}}.$$ In Equation (Equation 16), the form of the quasi-invariant limit depicts the evolution of wealth distribution.

### The Steady State Distribution of the Wealth

Rewriting Equation (Equation 16), we have
(17)$$\frac{\partial h^{*}\left(v,\tau \right)}{\partial \tau }=\frac{\lambda \kappa }{2}\frac{\partial^{2}\left[v^{\alpha +2}h^{*}\left(v,\tau \right)\right]}{\partial v^{2}}-\frac{\gamma \kappa }{2-\beta }\frac{\partial [\left(mv^{\alpha }log\frac{v}{v_{e}}-v^{\alpha +1}log\frac{v}{v_{e}}\right)h^{*}\left(v,\tau \right)]}{\partial v}.$$ The Fokker–Planck Equation (Equation 17) depicts the behavior of the scaled density function $h^{*}\left(v,\tau \right)$ for wealth *v*, where $v\in R^{+}$. As time approaches infinity, we have (18)$$\frac{\lambda }{2}\frac{\partial^{2}\left[v^{\alpha +2}h_{\infty }^{*}\left(v\right)\right]}{\partial v^{2}}-\frac{\gamma }{2-\beta }\frac{\partial [\left(mv^{\alpha }log\frac{v}{v_{e}}-v^{\alpha +1}log\frac{v}{v_{e}}\right)h_{\infty }^{*}\left(v\right)]}{\partial v}=0.$$ From Equation (Equation 18), we investigate the steady state of Equation (Equation 17). The stationary solution $h_{\infty }^{*}\left(v\right)$ of Equation (Equation 17) satisfies$$\frac{\partial h_{\infty }^{*}\left(v\right)}{\partial v}=-\frac{2}{\lambda v^{2}}\left[\frac{\lambda (\alpha +2)}{2}v-\frac{\gamma }{2-\beta }\left(mlog\frac{v}{v_{e}}-wlog\frac{v}{v_{e}}\right)\right]h_{\infty }^{*}\left(v\right),$$
which, in turn, has the stationary solution given by$$h_{\infty }^{*}\left(v\right)=\exp\left\{-\left(\alpha +2\right)logv-\frac{\gamma }{\lambda (2-\beta )}\left[log{\left(\frac{v}{v_{e}}\right)}^{2}-{\left(logv_{e}\right)}^{2}\right]-\frac{2\gamma m}{\lambda (2-\beta )}\left[\frac{log(\frac{v}{v_{e}})+1}{v}\right]\right\}.$$ Letting $\sigma_{s}=\frac{\lambda (2-\beta )}{2\gamma },\frac{1}{2\sigma_{s}}=\frac{\gamma }{\lambda (2-\beta )}$, we obtain the explicit steady-state distribution(19)$$h_{\infty }^{*}=C_{\sigma_{s},\alpha ,m}\cdot \frac{1}{v}\exp\left\{-\left[\frac{{\left(logv-\mu \right)}^{2}+\frac{2m}{v}\left(log\frac{v}{v_{e}}+1\right)}{2\sigma_{s}}\right]\right\},$$
where $\mu =logv_{e}-\sigma_{s}\left(1+\alpha \right)$, and $v_{e}$ is the essential wealth of the agent based on the intrinsic price of the stock. In particular, constant $C_{\sigma_{s},\alpha ,m}$ in Equation (Equation 19) depends on the average wealth *m*, the parameter of transaction frequency α, and $\sigma_{s}$ related to λ and β, satisfying$$\int_{0}^{\infty }h_{\infty }^{*}dv=1.$$

## 5. Numerical Simulations

This section presents an analysis of the steady-state profiles obtained from numerical simulations of the stationary wealth distribution. Basing on various key parameters, we observe how these profiles reflect the asymptotic behaviors of the steady-state solution to Equation (Equation 17), which is shown in Figure 2, Figure 3 and Figure 4.

Figure 2 demonstrates how the equilibrium wealth distribution evolves as the transaction frequency parameter α increases, while maintaining a fixed average wealth (m=0.15). The results depict a systematic shift in the distribution profiles, characterized by a migration of peak values toward the upper-left region. This pattern suggests that elevated transaction frequencies in the market lead to wealth concentration within specific clusters.

The relationship between average wealth *m* and steady-state wealth distribution is illustrated in Figure 3, where we observe the stationary wealth distribution across different values of μ as *m* varies from 0.05 to 0.2. The profiles reveal a consistent trend of wealth accumulation toward the lower-left region, accompanied by decreasing peak heights as *m* increases. Despite the broader wealth dispersion observed at a high mean wealth level, the distribution maintains characteristics consistent with empirical observations of wealth concentration among a small subset of market participants, which depicts a strong correspondence with the lognormal form of the wealth distribution, as approximated in the model.

In addition, we analyze the steady-state distribution $h_{\infty }^{*}$ for varying values of μ with average wealth *m* fixed though such profiles. In detail, the explicit stationary solution (19) indicates that the mean of the steady-state distribution, denoted as μ, is influenced by the parameter of transaction frequency α and the variance of the equilibrium distribution $\sigma_{s}$. The profiles in Figure 3 reveal that the vertices of each profile shift analogously toward the top-left corner as μ decreases, transitioning to a higher maximal point gradually. Since μ consequently diminishes as the transaction frequency increases, a similar mechanism is observed regarding the effect of transaction frequency.

Moreover, the stationary distribution is also affected by the parameter of psychological traits, denoted as β, introduced in the value function Ψ. To be specific, the underlying mechanism operates through the modulation of ratios between β, λ, and γ. Figure 4 illustrates the distribution behavior across different values of β, showing increased dispersion as the value of β rises. At that point, we obtain a clue that the condensation of the stationary wealth distribution is modulated through appropriate calibration of the psychological parameter, which reflects agents’ behavioral responses to the predicted and current prices of stocks.

Finally, these numerical results demonstrate the model’s capability to capture complex wealth distribution through the interaction of transaction frequencies, average wealth levels, and psychological traits. The findings align with empirical observations of wealth concentration patterns while providing insights into the mechanisms through which various parameters influence the distribution characteristics.

## 6. Conclusions

In this paper, we employ a kinetic exchange model based on microscopic binary interactions to analyze wealth distribution in a market characterized by financial assets (stocks and bonds). According to prospect theory, we introduce a value function that governs agents’ wealth allocation, reflecting their behavior in response to the predicted and current prices of the risky asset (stock). Additionally, our model considers the influence of transaction frequency on wealth distribution by incorporating a non-Maxwellian collision kernel. Applying the asymptotic limit method, we investigate the behaviors of wealth distribution, obtaining an entropy explicit stationary solution that exhibits exponential convergence to a lognormal wealth distribution. Numerical experiments are conducted on several parameters to support our theoretical results. 

## Figures and Tables

**Figure 1 entropy-27-00064-f001:**
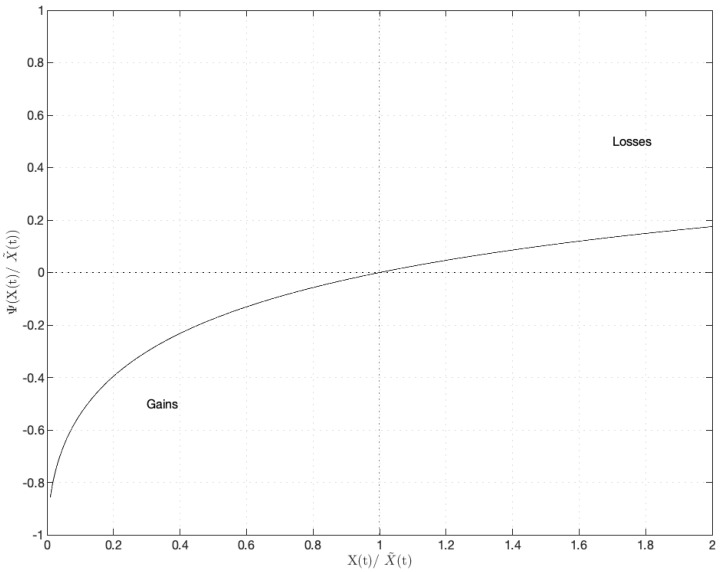
The illustration of value function $\Psi (\frac{X(t)}{\tilde{X}\left(t\right)})$.

**Figure 2 entropy-27-00064-f002:**
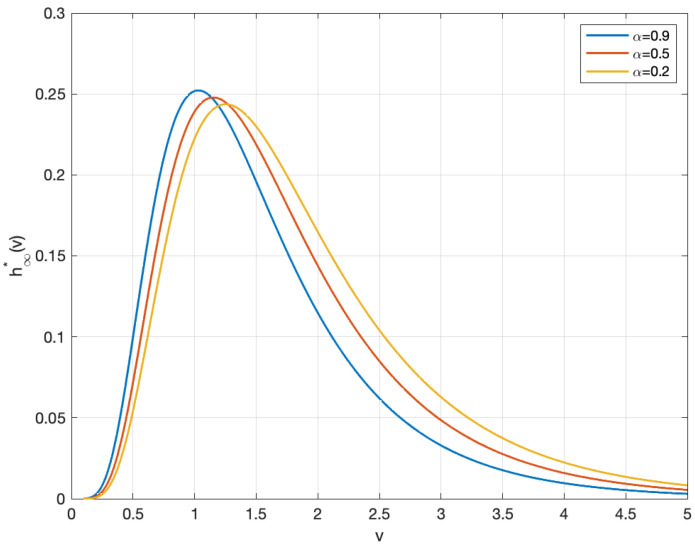
The stationary profiles of wealth distribution for m=0.15; α=0.9 (blue line); α=0.5 (red line); α=0.2 (yellow line).

**Figure 3 entropy-27-00064-f003:**
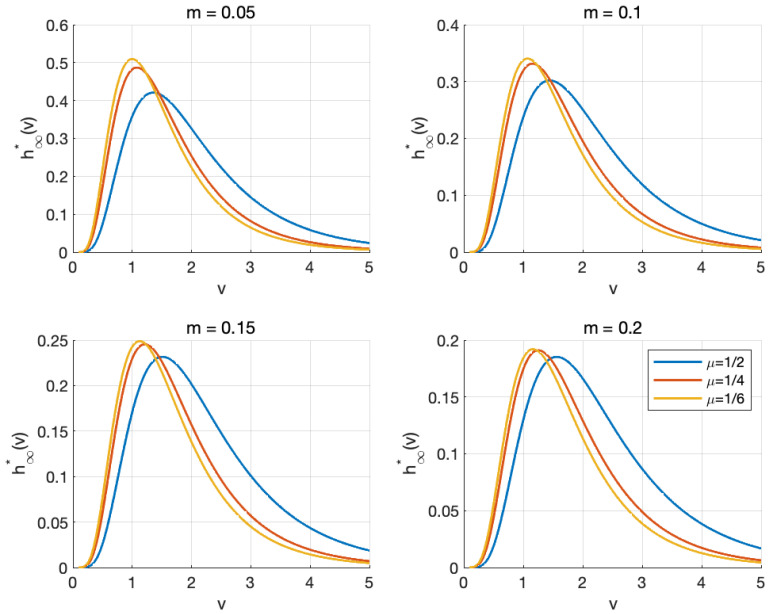
The stationary profiles for $\mu =\frac{1}{2}$ (blue line), $\mu =\frac{1}{4}$ (red line), and $\mu =\frac{1}{6}$ (yellow line) with different values of *m*.

**Figure 4 entropy-27-00064-f004:**
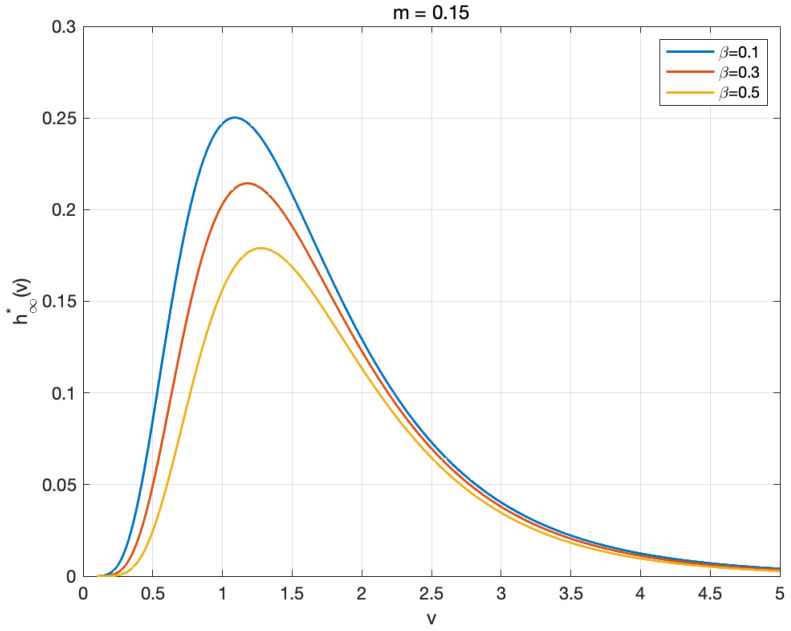
Profiles of the stationary solution $h_{\infty }^{*}$ for β=0.1 (blue line), β=0.3 (red line), and β=0.5 (yellow line) with m=0.15.

## Data Availability

No new data were generated or analyzed in this study; therefore, data sharing is not applicable.

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
