# Peer review of "Wealth Distribution Involving Psychological Traits and Non-Maxwellian Collision Kernel"

_entropy, 2025, doi:10.3390/e27010064_

Round 1
Reviewer 1 Report
Comments and Suggestions for Authors
In the submitted paper the authors discuss a novel model based on a model introduced by one of the authors in Ref. [11]. In the new model there is an additional parameter β in the utility function and they also consider a non-Maxwellian kernel. These are new features that affect the choices of the economic agents, a generalization which is in principle interesting and that could be discussed better, in relation to the old model and the literature.
As for the presentation, the article seems to be a line-by-line re-writing of the previous article [11], made starting from the very same text, adding the minimal changes (and some changes of notations).
This causes many repetitions that are probably unnecessary and pushes the new results at the very end of the paper.
In doing this, the authors don't pay enough attention to the new model when writing "We introduce a density function h(v, t), where w ⊆ R+ at the time t > 0, which satisfies..." (l.177) (there is no w variable in the paper, but w was the symbol for wealth used in Ref. [11]) - the same error is repeated in the following equation.
Fig. 1 is mysterious because it is not clear what it represents - for which parameters. The discussion of the parameter beta could have been done more in detail.
Writing a paper in this way is a risky strategy for writing a clear paper. So as it stands, the paper has some original results but I find that the paper can be greatly improved in readability, style, and also in the scientific discussion of the motivation of the model with respect to the rest of the literature.
Comments on the Quality of English LanguageOn the top of this, as a minor remark, the English of the text should be improved, since it contains many typos and expressions hard to understand. Here are some examples
45, making more realistic...
92 we discuss the affection of agents’ transaction frequency...
107 the kinetic model ... is enlightened by Zhong et al. [16]
111 the participation of the such assets
118 Following the clue in [11]...
133 to maximum their profit
133 We state the stock’s excess profit rate χ as follow..
.... there are other similar points to be re-checked in the rest of the paper
Reviewer 2 Report
Comments and Suggestions for Authors
Report on Article titled:
The Wealth Distribution Involving Psychological Traits and Non-Maxwellian Collision Kernel
Authors: Daixin Wang, Shaoyong Lai
Paper Entropy-3388752
The authors present an interesting approach to modeling wealth distribution in financial markets. By incorporating psychological traits of agents through a value function and introducing a non-Maxwellian collision kernel, the authors enrich the kinetic exchange model framework. The derivation of the Fokker-Planck equation and the exploration of steady-state wealth distributions are relevant for advancing the understanding of complex socioeconomic dynamics. The inclusion of numerical simulations further supports the theoretical presented framework.
In my opinion, the paper deserves to be published, however several points require clarification before it can be accepted for publication.
Specific Comments and Recommendations:
• The random variable $\eta_i$ introduced in Equation (5) is described as having zero mean and variance $\sigma^2$ , but I think, its role and generation mechanism within the model is not explained. Please, clarify whether this term is purely stochastic noise or if it represents a specific market phenomenon.
• The notation $\dot{X}(t)$ in numerator of equation not numbered in pg 3, between lines 126 and 127, presumably represents the time derivative of the price of the risky asset, is used without explicitly defining whether it is deterministic or stochastic. Given the inherent randomness of financial markets, it is more realistic to model $X(t)$ as a stochastic process. If $dot{X}(t)$ is intended to represent a deterministic rate of change, its relevance in a financial context should be better justified. Alternatively, if this is a typographical error, it should be corrected.
• The paper does not specify the number of agents used in the numerical simulations. This is an important detail for reproducibility and understanding the scale of the simulations. Please, include this information.
• The steady-state distributions obtained in the paper are log-normal, but power-law tails are a well-documented feature of wealth distributions in empirical studies. Discussing what possible modifications to the model could lead to power-law behavior would enhance the applicability and depth of the study.
• The model assumes homogeneous agents with identical parameters, which limits its realism. Discussing the implications of heterogeneity in agent behavior, such as differences in saving rates or psychological traits, could provide a better understanding of wealth dynamics.
• Clarification of Parameters $\beta$ , $\gamma$ and $\delta$. While these parameters are defined, providing a more intuitive explanation of their roles and possible empirical values would benefit readers less familiar with kinetic models.
• Figures presenting numerical simulations are clear but could be improved by including legends or annotations explaining key trends and parameter values.
• Proofread the manuscript for minor typographical issues, such as the notation discrepancies mentioned above.
In my opinion, this paper is a significant contribution to the field of kinetic modeling of wealth distribution. It combines innovative theoretical insights with practical simulations, making it an interesting and valuable study. However, before acceptance, the issues mentioned above must be addressed.
I recommend a minor revision to improve the clarity of the manuscript.

Reviewer 3 Report
Comments and Suggestions for Authors
In this paper the authors concentrate on a kinetic model to understand psychological phenomena in kinetic models for wealth distribution. The modelling aspects build upon Pareschi-Maldarella and Pareschi-Toscani papers, cited in the bibliography. The set-up is classical for this kind of problems. Anyway, I found a couple of issues:
1) in [13] it has been assumed that the evolution of the price in the profit rate \dot{X}/\mu, with \mu >0 the frequency of exchange rates. Now it is assumed \mu = 1, but the kinetic model has a kernel later on. How you relate this frequency with the introduced kernel at the kinetic level?
2) equation (2) is not fully justified, if we look at Pareschi-Maldarella [13] eq. 8 this is different. Please fully justify the approach.
3) P. 9 the quasi-invariant limit needs some regularity of the random variable \eta that is not included in the present approach. In particular, you classically need that the moment of order 2+\alpha is finite to justify that the reminder term vanishes, see Toscani '06 and Cordier-Pareschi-Toscani.
4) The numerical section is not really numerical. It would be worth including a test to show the convergence of the kinetic model to the Fokker-Planck steady-state for large times and \epsilon \to 0^+. Looking at the proposed plots of the steady states I do not find them very informative, I think that the computation of the Gini index would rather improve the modelling understanding of this paper.
Round 2
Reviewer 1 Report
Comments and Suggestions for Authors
The new version submitted has been improved with respect to the original submission.
Besides correcting the typos, the authors have added more details, information, explanations, which make the paper more readable, consistent, and interesting.
I find the paper suitable to publication in the present format.
As a last suggestion, it would be good to add some more comments and elaborate on the non-Maxwellian form of the collisional dynamics, in particular its relation with other models of trade dynamics or different types of models used in modelling trades, since this an interesting and relevant point of the paper.
Reviewer 3 Report
Comments and Suggestions for Authors
The authors answered to my concerns and I suggest acceptance of the paper in its present form.
